

**Altitudinal distribution of soil organic and inorganic carbon in a dry**
**alpine rangeland of northern Qinghai-Tibetan Plateau**
Qinglin Liu [a], Ailin Zhang [a*], Xiangyi Li [b], Jinfei Yin [a], Yuxue Zhang [a], Osbert Jianxin
Sun [a,c], Yong Jiang [a]
[a] School of Life Sciences, Hebei University, Baoding, 071002, China
[b] Xinjiang Institute of Ecology and Geography, Chinese Academy of Sciences, Urumqi, 830011,
China
[c] School of Ecology and Nature Conservation, Beijing Forestry University, Beijing, 100083, China
* Corresponding authors.
E-mail address: alzhang@hbu.edu.cn (Ailin Zhang).
**Abstract**
The spatial patterns of soil carbon in water-constrained alpine ecosystems have been
rarely investigated. It remains unclear how changes in biotic and abiotic factors with
altitude would shape the distribution of soil carbon stocks when plant communities
are co-limited by water and low temperature. To address this uncertainty, we
investigated changes in soil organic carbon (SOC) and inorganic carbon (SIC) along
an altitudinal gradient between 3000-4000 m asl, in the northern Qinghai-Tibetan
Plateau. Our results showed that the total soil carbon density (TCD) and the SOC
density (SOCD) increased with increases in altitude, but the SIC density (SICD)
displayed a pattern of nonlinear change along the altitudinal gradient with a peak at
the mid-slope of the range. While SIC dominated the soil carbon pool, accounting for
64 - 90% of TCD, the proportion of SOC increased from 10 to 36% of the TCD with
increases in altitude. The increases in SOCD with altitude were associated with
changes from scrub-dominated vegetation cover to herbaceous plant communities and
decreasing MAT, which together attributed to increased level of plant-derived carbon
inputs and reduced SOC mineralization at higher altitudes. Whereas variations in
SICD were mainly explainable by changes in soil C/N and soil water content (SWC),
and likely resulted from non-linear changes in factors related to inorganic carbon



production and leaking losses. Findings from this study help fill the knowledge gap on
the underlying controls of SOC and SIC distribution along the altitudinal gradient in
water- and low temperature-constrained alpine rangeland.
**Keywords** Qinghai-Tibet Plateau · arid region · C pool · soil organic C · soil
inorganic C
**1.  Introduction**
Because soil contains the largest proportion of C stocks in terrestrial ecosystems
(Lal, 2018), the size, persistence and storage capacity of soil C pool have been the
focal issues in global change research. However, despite extensive studies, there are
still great uncertainties in the response to, and mitigation potential of, global climate
change by soil C. Part of the problems arises from differential alterations of pool size
and functional structure of soil C among the world's terrestrial ecosystems as affected
by environmental variability and climate change (Sun et al., 2019, 2023; Zhang et al.,
2024). Soil C pool is made up of both organic (SOC) and inorganic chemical
compounds (SIC). In general, SOC dominates the soil C pool on vegetated sites
(Feyissa et al., 2023); whereas SIC is a major component of soil carbon pool in arid
areas where plants are scarce especially in drylands (Du and Gao, 2020; Dong et al.,
2024). Previous research has well demonstrated that SOC are jointly controlled by
vegetation, climate and soil physicochemical properties (Eswaran et al., 1993; Torn et
al., 1997; Schuur et al., 2001; Callesen et al., 2003; Sun et al., 2004). In contrast, SIC
pool is mainly affected by abiotic factors such as soil parent material, climate and
altitudes gradient (Chang et al., 2012; Ma et al., 2022; Dong et al., 2024). Under
different altitude gradients and vegetation types, the change rules of SOC and SIC
pools and the specific response factors are still unclear.
The Qinghai-Tibetan Plateau is characterized by a drastic rise in elevation and
occurrence of spatial divergence in ecosystem types typically exhibited by latitudinal
and longitudinal patterns of vegetation but within a much-confined space. The unique
topographic feature and presence of diverse ecosystems have made the region a hot
spot for research geared at better understanding of the impacts by climate change on





ecosystem structure and function. However, most of the studies in the region have
been conducted with primary objectives to elucidate the changes and underlying
controls in wetland and grassland plant communities, and/or changes in soil properties,
in relation to the regional trend of climate change and permafrost degradation (Wang
et al., 2023a; Chen et al., 2017; Cai et al., 2025), with the alpine dry rangeland been
largely neglected. On the Qinghai-Tibetan Plateau, the abiotic conditions, vegetation
and soil types have undergone great changes at different elevations (Li et al., 2017),
which would inevitably impose significant impacts on soil C dynamics due to
altitudinal shift in vegetation type and hydrothermal conditions (Rodeghiero and
Cescatti, 2005). Previous studies have shown substantial variations in the quality and
quantity of SOC along the altitudinal gradients in mountainous landscapes (Pepin et al,
2015). However, in arid alpine rangelands, such as that in the northern
Qinghai-Tibetan Plateau region, vegetation cover is most sparse and SIC plays a more
dominant role the soil C storage (Batjes, 2006; Du and Gao, 2020). Therefore,
previous studies have neglected the role of SIC in soil carbon inventory in terrestrial
ecosystems. The Qinghai-Tibetan Plateau is a climate-sensitive region and the
altitudinal variations in climatic factors are more pronounced because of the sharply
raised terrain (You et al., 2021). With increases in altitude along the mountain slopes
on the Plateau, air temperature markedly decreases, and precipitation and the intensity
of solar radiation increases, contributing to altitudinal changes in vegetation and soil
nutrient availability (Tiemann and Billings, 2011; García-Palacios et al., 2013; Wang
et al., 2023b). But it is not known if and to what extent the altitudinal changes in
micro-environments and vegetation would affect soil C and if the stocks of SOC and
SIC would co-vary with altitude.
In this study, we investigated the patterns of changes in SOC and SIC in the top
30 cm soil along the southern slope of the Altun Mountain across an altitudinal range
of 3000 - 4000 m above sea level (asl), in the northern Qinghai-Tibetan Plateau region,
and collected data on plant communities and soil physicochemical properties. The
aims of the study were to determine the altitudinal patterns in the density of SOC





(SOCD) and SIC (SICD). We hypothesized that (1) the relative importance of SOC to
SIC decreases with altitude because of the imbalance between the inputs and
mineralization of soil organic matter, and (2) SICD would become more profound at
higher altitudes because of reduced inputs of plant-derived organic matter.
**2.  Methods and materials**
2.1 Study sites and experimental design
Our study sites are located in the Altun Mountain Nature Reserve in the south of
Altyn Tagh, situated in the northeastern part of the Qinghai-Tibetan Plateau (87°10'E -
91°18'E, 36°N - 37°49'N). This area is known for harsh environmental conditions
characterized by a dry climate, with an average annual temperature of 0 °C and an
annual precipitation of around 110 mm. The soils are predominantly yermosols (FAO,
http://www.fao.org/soils-portal/so). The Reserve comprises diverse landcover types,
including deserts, scrubs, and grasslands. The herbaceous layer typically ranges from
5 to 20 cm in height, with a coverage of 10 - 30%, occasionally reaching 60 - 80%.
The main vegetation types being dwarf scrubs in the lower altitudinal range of the
slope and grassland at the upper slope. In this study area, the vegetation types are
mainly small shrubs and shrubs at the altitude of 3000-3300m, the vegetation types
are mixed shrubs and herbs at the altitude of 3300-3500m, and the vegetation types
above 3500m are mainly herbs (Fig S1). Dominant plant species are represented by
*Stipa purpurea* Griseb. and *Kobresia robusta* Maxim., which are often accompanied
by common grassland plants including *Carex kunlumsannsis* N.R.Cui, *Koeleria*
*cristata* (L.) Pers., and *Oxytropis falcata* Bunge.
In August 2019, we conducted plant survey and soil sampling at seven altitudes
(designated as A1-A7) along a vertical transect in an altitudinal range of 3000-4000 m
asl in the northern section of the Altun Mountain Nature Reserve. The distance
between adjacent survey and sampling altitudes ranged from 60 to > 100 m in
elevation. All sites are geo-referenced. At each altitude, five quadrats (each measuring
$100 \times 100$ cm) were setup along the contour for measurements of plants and soil



sample collections, and the quadrats were separated by a ~20 m spacing between
adjacent ones. The altitudinal profile of the sampling sites is illustrated in Fig. 1.

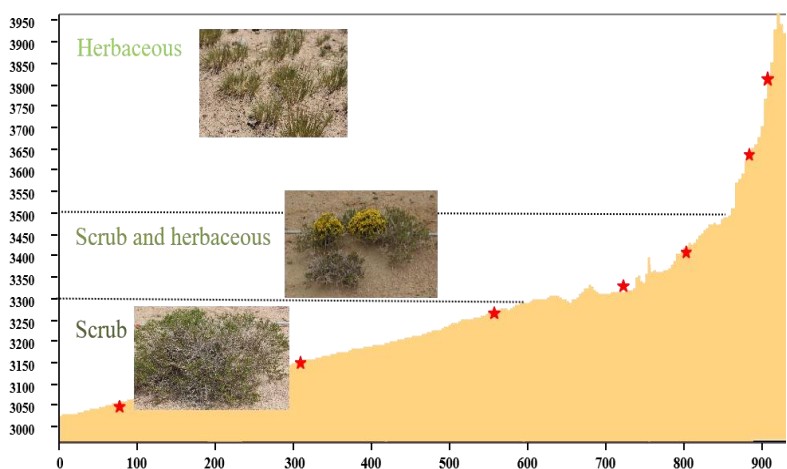


Fig. 1 The altitudinal profile of sample sites in the northern section of the Altun

Mountain Nature Reserve, Qinghai-Tibetan Plateau

2.2 Measurements of plant and soil variables

We determined the relative coverage of each species and the entire community

and identified the dominant species. The coverage of the plant community was
calculated as the sum of the coverage values for individual species as there are little
overlap among plant species at our study sites. All plants within each quadrat were
harvested and measured for both fresh and dry mass. Soil samples were collected to
30 cm depth using a 7-cm (inner diameter) augur at locations where the above-ground
tissues were harvested for biomass measurements. Roots were picked out of soil
samples and measured for dry mass weight. Upon completing the field survey, plant
samples were transported to laboratory and oven-dried at 75 °C for 48 h for
determination of biomass. Soil water content (SWC) was determined gravimetrically
by determining the fresh soil weight and then dry mass after subjecting to oven-drying
at 105 °C for 48 h. Soil pH was determined using a conductometer (1:1 soil-water
suspension) and acidimeter (1:5 soil-water suspension). Soil bulk density (BD) was
determined using cutting ring method. The SOC and plant C contents (above-ground

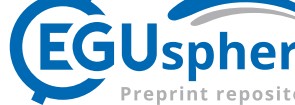



tissues and roots) were quantified using the $K_2Cr_2O_7$ oxidation, and soil TC using an
elemental analyzer (TOCV wp; Shimadzu Corp., Tokyo, Japan). The SIC content was
determined through neutralization titration. The measurements for plant C were made
with samples oven-dried at 75 °C for 48 h.
2.3 Data processing and statistical analysis

The climate data (MAT, NDVI) used in this study were extracted from the

website of Global climate data (http://worldclim.org). For plant community structure,
we quantified Shannon-Wiener index ($H'$) and species richness (Whittaker and
Niering, 1965).

One-way ANOVA was used to determine the effects of altitudes on SWC and BD,

and Duncan's multiple comparison test to determine the statistical significance of the
differences of the variables among altitudes. Linear regression was applied to examine
the relationships of SOC and SIC with the indices of climate, plant community and
soil. Redundancy analysis (RDA) was performed for identification of significant
influencing factors on SOC and SIC, and Structural Equation Modelling (SEM) for
analysis of direct and indirect influences of climate, plant and soil variables on SOC
and SIC. We adjusted the model according to the theoretical understanding of the
processes and removed the paths that are not significant or have only weak effects.
Data were fitted to the models using the maximum likelihood estimation method
(Tian et al., 2021). All statistical analyses were implemented within Origin 9.3 and R
4.2.1 (R Core Team, 2020), and SEM-composite analysis in the R packages
"piecewise SEM", "nlme" and "lme4".
**3. Results**
3.1. Changes of soil C pools, plant community, climate and edaphic factors with
altitudes

The total soil C (TC) pool was predominantly made up of inorganic component

along the altitudinal gradient, accounting for 64 - 90% of TC (Fig. 2). With increases
in altitude, the organic component of soil C (SOC) significantly and linearly increased
(Fig. 2a), leading to its increased proportion from 10 to 36% of TC (Fig. 2b).



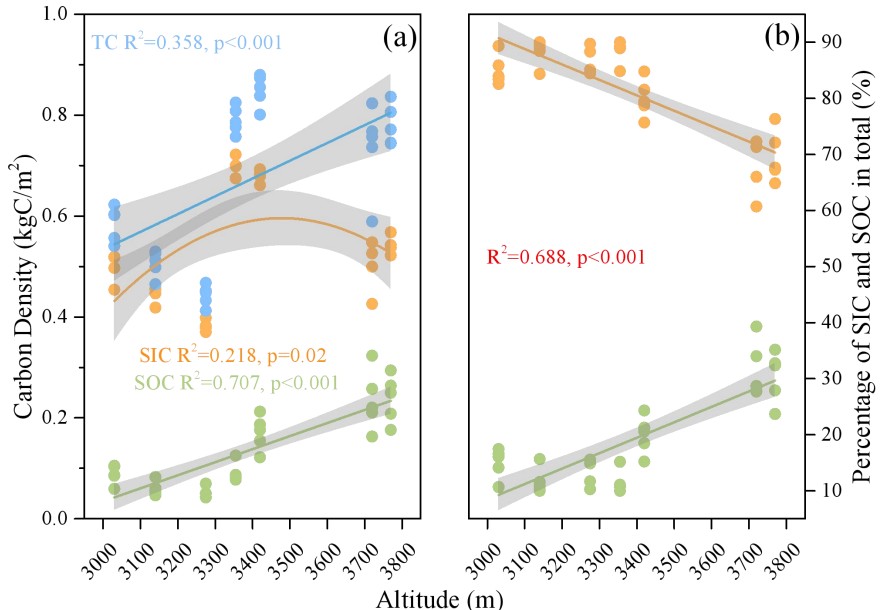

Fig. 2 Altitudinal changes in (a) densities of soil organic (SOC), inorganic (SIC) and total C (TC), and (b) proportions of SOC and SIC in TC in a dry alpine rangeland of Qinghai-Tibetan Plateau

Among the variables characterizing plant communities, NDVI, plant species diversity and richness all increased with rising altitude ($p<0.01$) (Fig. 3a-c), and above-ground biomass density decreased ($p<0.01$) (Fig. 3e). However, both fine root biomass density and vegetation cover displayed patterns of curvilinear changes with altitudes (Fig. 3d and 3f).

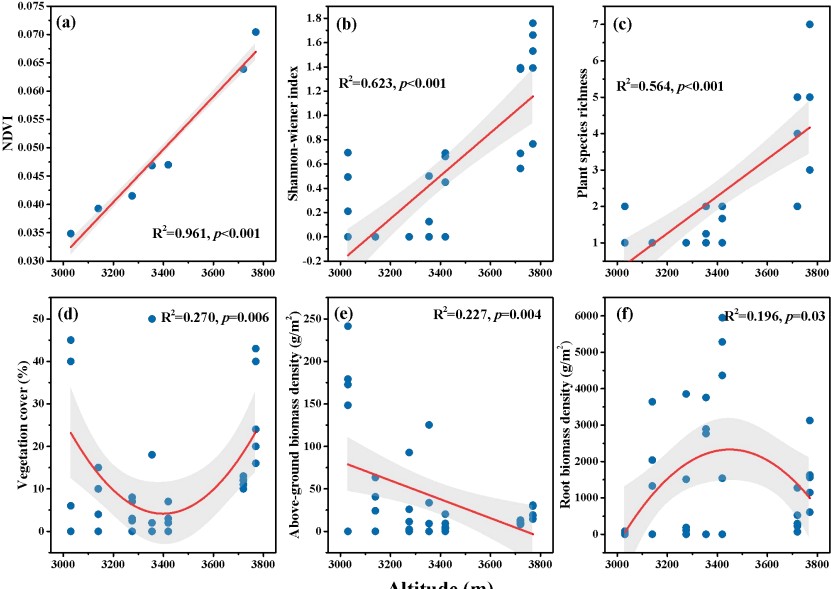


Fig. 3 Altitudinal changes in selective plant community traits in a dry alpine
rangeland of Qinghai-Tibetan Plateau. (a) Normalized difference vegetation index
(NDVI); (b) Shannon-Wiener index; (c) plant species richness; (d) vegetation cover;
(e) aboveground biomass density; and (f) fine root biomass density
Among the climatic and edaphic variables, there were a significant linear
decrease in MAT ($p<0.01$; Fig. 4b) and a significant linear increase in soil N content
increased with increases in altitude ($p<0.01$; Fig. 4d). Soil pH, BD and C/N all
exhibited a hump-shaped pattern of changes along the altitudinal gradient ($p<0.01$;
Fig. 4c, e, f).

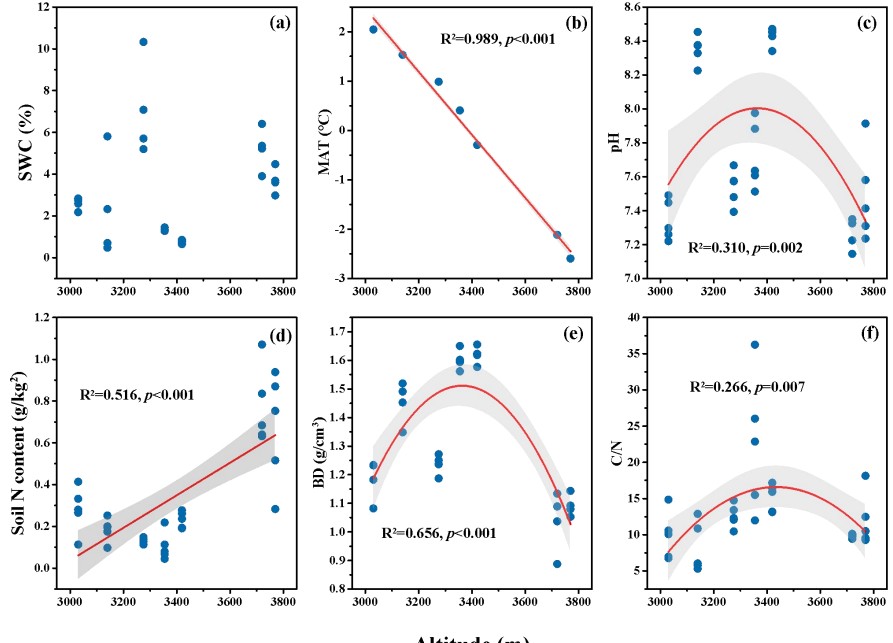

Fig. 4 Altitudinal changes in selective climatic and soil variables in a dry alpine

rangeland of Qinghai-Tibetan Plateau. (a) Soil water content (SWC); (b) mean annual

temperature (MAT); (c) soil pH; (d) soil N content; (e) soil bulk density (BD); (f) soil

C to N ratio (C/N)

3.2. Influencing factors on changes in soil organic C density (SOCD) and inorganic C
density (SICD)

The results from both correlation analysis and RDA showed that SOCD had a
significant negative correlation with MAT, and significant positive correlations with
BD, soil N content, NDVI and plant diversity (Table 1; Fig. 5). In contrast, SICD was
negatively correlated with SWC, and positively with BD, soil C/N and fine root
biomass density (Table 1; Fig. 5). In general, changes in SOCD was mostly
explainable by variables related to climate and plant community traits; whereas SICD
was predominantly associated with edaphic factors.




Table 1 Summary of the correlation coefficients for relationships of SOCD and SICD with
selective variables for climatic, edaphic and plant community characteristics. * *p*<0.05, ***p*<0.01

| | Climatic factors | | | Soil factors | | | | Plant community factors | | | | | |
|---|---|---|---|---|---|---|---|---|---|---|---|---|---|
| | SWC | MAT | pH | BD (g/cm$^3$) | Soil N content | C/N | NDVI | Shannon-wiener index | Plant species richness | Vegetation cover | Above-ground biomass | Root biomass |
| SOCD | 0.039 | -0.872** | -0.244 | 0.394* | 0.885** | -0.128 | 0.843** | 0.843** | 0.771** | 0.279 | -0.265 | 0.12 |
| SICD | -0.718** | -0.224 | 0.311 | 0.634** | -0.09 | 0.561** | 0.163 | 0.113 | 0.055 | 0.034 | -0.093 | 0.412* |


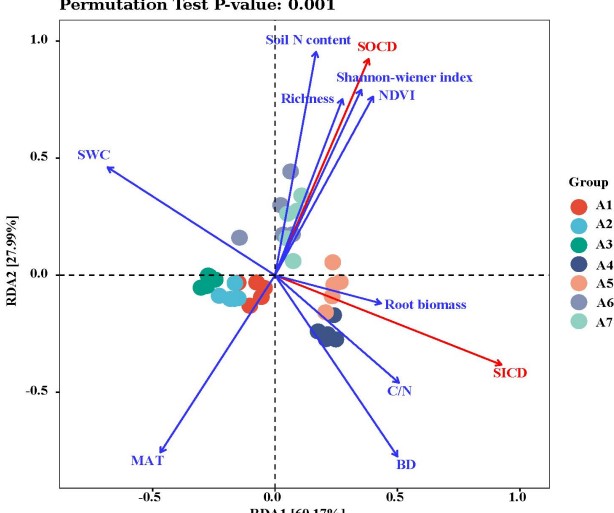


Fig. 5 RDA ranking of soil C pool (red line) and environmental variables (blue line)
at different altitudes. Arrow-lines represent relative values of environmental variables
and soil C pool. Correlations between environmental variables and soil C pool are
indicated by the cosine of angles between the corresponding arrow-lines; angles <90°
indicate a positive correlation, and >90° a negative correlation. Projecting the
arrow-line for a soil C pool into an arrow-line for a corresponding environmental



variable, the distance from the origin to the projection point indicates the relative
power of the environmental variable in explaining the size of soil C pools.
In the structural equation modelling, the effects of altitude on SOCD were
implemented via modifications of climate and plant communities (Fig. 6a); whereas
variations in SICD were mainly associated with edaphic factors (Fig. 6b), consistent
to the results from the RDA.

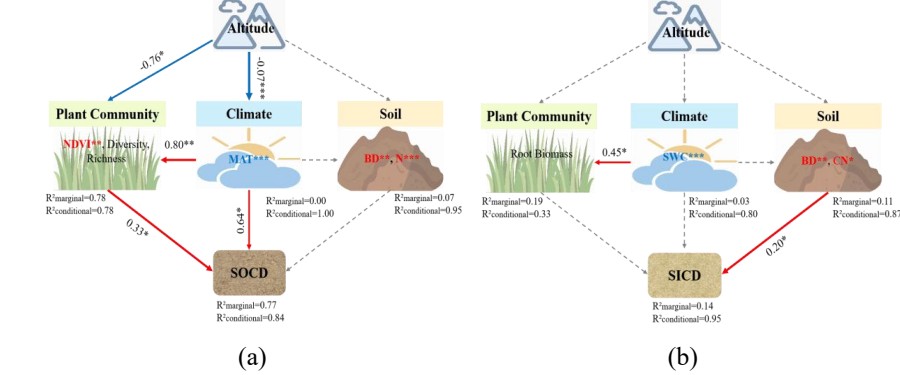


(a)                                          (b)
Fig. 6 Structural equation models of the influences on (a) soil organic C density
(SOCD) and (b) soil inorganic C density (SICD) by altitude, climate, and plant
community, and soil. (a) Fisher's C = 4.714; p = 0.318; df = 4; AIC = 44.714; BIC =
75.821; (b) Fisher's C =5.000; p = 0.287; df = 4; AIC = 45.000; BIC = 76.107.
Numbers adjacent to arrows are the standardized path coefficients (equivalent to
correlation coefficients). Arrow thickness indicate the strength of the relationships.
Red solid arrows denote significant positive effects (p<0.05) or marginally significant
(0.05<p<0.1) effects. Blue solid arrows denote significant negative effects (p< 0.05)
or marginally significant (0.05<p<0.1) effects. $R^2$ values associated with response
variables indicate the variance accounted for by the model. * $p < 0.05$; ** $p < 0.01$;
*** $p < 0.001$.





**4.Discussion**

Soil carbon pool, as the largest carbon pool in terrestrial ecosystem, has been extensively studied in different scales and regions (Zhang et al., 2024; Chalchissa and Kuris, 2024). However, previous studies have not paid much attention to the composition of different carbon pools in extreme environments. We studied the altitudinal patterns of soil organic and inorganic C pools in an alpine rangeland where ecosystem processes are co-limited by drought and low temperature. In contrast to the findings from prior studies with alpine meadow or moist grasslands (Chen et al., 2017; Chen et al., 2022), our results show the predominance of soil inorganic C in the dry alpine rangeland. Our results show a linear increase in soil organic C pool with rising altitude. However, the pattern of changes in soil inorganic C pool appears to be nonlinear along the altitudinal gradient.

In this study, the linear changes in soil organic C pool along the altitudinal gradient were positively related to the altitudinal distribution of plant diversity and NDVI, but negatively to aboveground biomass density. It is generally found that increases in plant diversity and species richness promote the formation of soil organic C (Gu et al., 2019; Xu et al., 2021; Spohn et al., 2023). This is because SOC are predominantly derived from plant residues (Schmidt et al., 2011). More diverse plant species optimize the complementary use of resources and increase community productivity in areas with lower species richness (Lehmann et al., 2020). The negative correlation between soil organic C pool and aboveground biomass density in this study can be explained by the shift in vegetation cover type from slow-turnover scrubs (e.g. *Krascheninnikovia compacta* (Losinsk.) Grubov and *Salsola abrotanoides*) at the lower altitudinal range to fast-turnover grassland plants (e.g. *S. purpurea* and *P. bifurca*) at the higher altitudinal range. Scrubs typically have greater standing biomass but much slower turnover rate the organs and tissues than herbaceous plants. Moreover, with increases in altitude, temperature decreased and precipitation increased, both of which favoring the preservation of soil organic C. Therefore changes in both vegetation and climatic conditions led to an increased SOC pool



content at the higher altitudes (De Deyn et al., 2008).

Climate is an important abiotic factor affecting the size and stability of soil C

pool (Possinger et al., 2021; Zhang et al., 2024). Our study shows that decreases of
MAT and SWC contributed to increased SOCD and SICD. This is contrary to the
findings from previous studies in humid environments that SOC increases with rising
temperature (Chalchissa and Kuris, 2024; Jiang et al., 2024). The discrepancy is
mainly because that our study area is situated in an extremely arid region, such that
the transpiration effect is much greater than that of precipitation. When temperature
decreases, the transpiration effect decreases significantly, which is more conducive to
plant growth and soil C accrual (Schmidt et al., 2011). In addition, lower temperature
also significantly inhibits the activity of soil microorganisms, reducing the microbial
decomposition of soil organic matter (Sun et al., 2019). In the case of small climate
differences brought about by changes in altitude gradient, vegetation abundance and
diversity increase with the increase of altitude gradient, which leads to the increase of
plant carbon input, but the mineralization of microorganisms remains unchanged (Yue
et al., 2017).

Apart from the effects of climatic factors and plant community factors, previous

studies also suggested that soil properties had direct and major effects on soil C stock
(Hemingway et al., 2019). In this study, we found that soil N level was greater at the
higher altitudes, favoring the accumulation of SOC (Puspok et al., 2023). This is
mainly because that an increase in soil N content suggests the greater abundance of
nitrogen-fixing plants and/or microorganisms, and the acceleration of underground N
cycling; under which conditions plants grow faster and turn over more rapidly,
thereby enhancing the inputs of soil organic matter (Reay et al., 2008; Sonam et al.,
2016). Overall, however, we found that the effect of soil factors on SOC was weak,
and indirectly through biological factors in the form of plant community structure (Fig.
6a).

In contrast to the clear altitudinal pattern of SOCD, SICD did not display a

consistent pattern of altitudinal changes. It initially decline with altitude up to about





3300 m asl (Fig. 2), but peaked at about 3400 m asl, at a position where the gentle
slope at the lower altitude gave way to a much steeper mountain slope (Fig. 1). The
abnormally high value of SICD at the foot of steep mountain slope could be
consequences of alluvial deposit and accumulation of carbonate salt originated from
the uphill, hence a reflection of geological and hydrological effects. The correlation
analyses revealed that SICD was mostly related to non-biotic factors such as soil bulk
density and soil water. The main constituents of the SIC reservoir are carbonate salts
(Zhao et al., 2019). When soil water content is high, $CO_2$ is readily transformed into
carbonic acid ($H_2CO_3$), carbonate ($CO_3^{2-}$), and bicarbonate ($HCO^{3-}$), which promotes
the dissolution of calcium carbonate and reduces the SIC content (Huber et al., 2019).
The greater precipitation at the higher altitudes may facilitate the leaching of SIC to
the deep layer, resulting in a decrease in the surface soil C pool (Du and Gao, 2020).
As a result, soil SIC content is higher and more stable in the arid soils of high
elevation (Ren et al., 2024). Previous studies have shown that SIC is not only affected
by abiotic factors, but also by biological factors (Ma et al., 2024). Increased plant
growth and biological activity enhance root respiratory secretion, resulting in
dissolution and loss of SIC (Kuzyakov and Razavi, 2019). In this study, however,
SICD was weakly correlated with root biomass, likely due to the low level of soil
development in the study area. In the SEM-composite model analysis, the increases in
BD and soil C/N led increased level of soil inorganic C (Fig. 6b). There are two main
reasons for this phenomenon: on one hand, the distribution law of SIC comes from the
distribution law of soil parent material itself, which has nothing to do with external
factors; on the other hand, the turnover rate of SIC is low, the time stability is high,
and the response to influencing factors is weak. The change of SIC in arid area is
mainly influenced by long-term geological cycle.
**4. Conclusion**
Contrary to our both hypotheses, our results show increased SOC with altitude
while not exhibiting a clear directional change in SIC in the dry alpine rangeland of
Qinghai-Tibetan Plateau. SOC increased its relative contribution to total C pool at the



higher altitudes because of changes in plant communities and climatic conditions
promoting soil organic C production and preservation. Overall, inorganic C played a
predominant role in determining the soil C pool size in dry alpine rangeland, the
difference of SIC distribution at different altitudes is not affected by vegetation and
climate change caused by altitude gradient. The results show that the soil carbon pool
in the alpine desert region is mainly composed of SIC compared with that in the
humid region, but the influence of climate, vegetation and other environmental
conditions on the soil carbon pool is mainly achieved by changing SOC. Therefore,
maintaining ecological stability in cold and dry region has an important impact on the
carbon cycle of terrestrial ecosystems.

*Data availability.* The links to data are provided in the paper.

*Authorship contribution.* ALZ conceived and designed the experiments; QLL wrote
the manuscript; JFY and YXZ analysed the data and contributed to the discussion;
XYL, OJS, and YJ revised the article.

*Competing interests.* The authors declare that they have no known competing
financial interests or personal relationships that could have appeared to influence the
work reported in this paper.

*Disclaimer.* Publisher's note: Copernicus Publications remains neutral with regard to
jurisdictional claims made in the text, published maps, institutional affiliations, or any
other geographical representation in this paper. While Copernicus Publications makes
every effort to include appropriate place names, the final responsibility lies with the
authors.

*Acknowledgments.* We thank all participants for contribution to the field work and



laboratory analysis.

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
