# Peer review of "Changes in soil organic and inorganic carbon with elevation in a dry alpine rangeland of northern Qinghai-Tibetan Plateau"

_EGUsphere, 2025_

## Author Comment (AC1)

**Reviewer #1:**

Thank you very much for the reviewers' suggestions. We have made the following modifications to the questions raised by the reviewers:

The manuscript by Liu et al. investigated the pattern of changes in soil organic carbon and soil inorganic carbon along the 3000-4000 m altitude range in the northern Qinghai-Tibetan Plateau region. In this study, SOCD increased with the elevation of altitude and was attributed to the changes of dominated vegetation cover and the decreasing MAT. SICD displayed a nonlinear change and was influenced by soil C/N ratio and soil water content. The topic of the study is very interesting and they found the divergent responses of soil C component to altitude. However, there still some issues needed to be addressed before its publication. First, it is important for the authors to present scientific questions more clearly based on summarizing previous research, and further improve the logical and English expression of the paper. Second, the figure and table notes in the result section need to be modified according to the format and need to be distinguished from the font in the manuscript. Third, in the discussion section, there is still a problem of logic ambiguity in the change of soil inorganic carbon density. Please re-organize the structure of this section, so as to make the logic of the discussion section more reasonable and clearly. Further, there are many long and complex sentences in the article, please modify wherever needed. I also have some specific comments listed below.

Response: Based on the reviewers' comments, we further clarified the assumptions, corrected the chart errors, and reorganized the analysis of SIC. The specific modification contents can be found in the main text of the article and specific comments.

Specific comments:

Line 15 Please describe the experimental design in detail

Response: In the abstract, we have added the description of the experimental design. "To address this uncertainty, we uniformly selected seven points on the vertical gradient at an altitude of 3,000-4,000 meters in the northern part of the Qinghai-Tibet

Plateau to analyze the changing trends of organic carbon (SOC) and inorganic carbon (SIC) in the surface soil."

Line 27 Please add the "ratio" after C/N

Response: Thanks very much to the suggestions of the reviewers, we have revised the sentence.

Line 32 Don't use abbreviations when writing keywords and try to spell the words in full.

Response: Thanks very much to the suggestions of the reviewers, we have revised the keywords.

Line 37 The meaning of the sentence is not clear, please revise it.

Response: We modify this sentence to "However, despite extensive studies on soil carbon pools in the past, there is still great uncertainty regarding the response of soil C pools to global climate change'.

Line 44: Please revised "whereas" to "Whereas"

Response: Thanks very much to the suggestions of the reviewers, we have revised the sentence.

Line 52 the driven factors?

Response: Based on the reviewers' opinions, we added specific influencing factors such as climate and vegetation. For specific modifications, please refer to line 52-54.

Line 53-55 This sentence is long, so please split into two short sentences.

Response: We modify this sentence to "The Qinghai-Tibet Plateau rises sharply in the altitude gradient, and the ecosystem types have the characteristics of spatial differentiation. Among different ecological types, there are significant differences in climate, vegetation and soil characteristics".

Line 58-63 Again, the description of research progress is also too long. It will be better to raise the knowledge gap in a separate sentence.

Response: According to the reviewers' comments, we modify this sentence to " However, most studies in this region have mainly focused on the vegetation or soil change patterns of permafrost and meadow ecosystems (Wang et al., 2023a; Chen et al., 2017; Cai et al., 2025. The variation patterns of vegetation and soil carbon pools

in alpine meadow ecosystems in arid areas have been largely ignored".

Line 70 delete most

Response: Thanks very much to the suggestions of the reviewers, we have revised the sentence.

Line 71 in the soil C storage

Response: Thanks very much to the suggestions of the reviewers, we have revised the sentence.

Line 72 Previous studies may have neglected the role of SIC in soil carbon inventory in this region.

Response: Thanks very much to the suggestions of the reviewers, we added this sentence.

Line 73-75 Please delete this repetitive sentence.

Response: Thanks very much to the suggestions of the reviewers, we have revised the sentence.

Line 75-78 These statement about the altitude changes in vegetation and soil nutrient availability should have appeared earlier in this paragraph.

Response: We placed the description of the changes in vegetation and nutrients on the altitude gradient in paragraphs line 58-62.

Line 79 whether or to what extent the altitudinal changes in micro-environments and vegetation would affect SOC and SIC remains unknown.

Response: Thanks very much to the suggestions of the reviewers, we have revised the sentence.

Line 87-90 The two hypotheses are similar: the relative importance of SOC to SIC decreases with altitude vs. SIC would become more profound at higher altitudes.

Response: According to the reviewers' comments, we modified the hypothesis as "(1) With the increase of altitude, the decrease of temperature and the increase of vegetation, the content of soil organic carbon increases; (2) With the increase of altitude, the drought limitation is alleviated and the inorganic carbon content in the soil decreases; (3) With the increase of altitude, the proportion of organic carbon in the soil carbon pool increases, while the proportion of inorganic carbon decreases."

Line 115 The description of the methods that the experiment was conducted should be further addressed.

Response: Thanks very much to the suggestions of the reviewers, we have revised the methods in line 118-121.

Line 126 The soil was collected from 0-30 cm, which was different from 0-20 cm in other literatures. Please explain in the materials and methods and explain why this soil layer is used.

Response: This is mainly because in this study area, there are more vegetation such as dwarf shrubs, and the root systems are mostly distributed in the range of 0-30cm. Therefore, we chose a soil layer of 0-30cm.

Line 136 The measuring method of SIC is unclear. Is it a calculated value by subtracting SOC from TC or a measured value?

Response: The content of SIC was directly determined by neutralization titration.

Fig. 2b There were two regression lines, but only one R2 and P values was reported?

Line 198 In Table 1 and Figs 3, 4, the listed variables mismatched, please revise them in the result.

Response: We have corrected the errors in the chart. For the specific correction content, please refer to the chart in the main text.

Line 235: showed

Response: Thanks very much to the suggestions of the reviewers, we have revised the sentence.

Line 236-238: Further, soil organic C pool linearly increased, while soil inorganic C pool appears to be nonlinear along the altitudinal gradient.

Response: Thanks very much to the suggestions of the reviewers, we have added the sentence in line 246-247.

Line 246 Unexpectedly, this study found the negative correlation between soil organic C pool and aboveground biomass density. This could be explained by…

Response: Thanks very much to the suggestions of the reviewers, we have added the sentence in line 255-257.

Line 258 showed

Response: Thanks very much to the suggestions of the reviewers, we have revised the sentence.

Line 263, 264 delete "effect"

Response: Thanks very much to the suggestions of the reviewers, we have revised the sentence.

Line 267-271 Please make a summarized conclusion about how vegetation abundance and climate together affect SOC here.

Response: According to the research content, a summary of the impact on SOC has been added at the end of this paragraph. "Therefore, with the increase of the altitude gradient and the decrease of temperature, the vegetation type gradually changes from desert to grassland, the soil carbon input increases, the mineralization rate decreases, and the SOC content increases."

Line 278 Under this condition, plants may grow faster…

Response: Thanks very much to the suggestions of the reviewers, we have added the sentence in line 291.

Line 280-282 Please move this main result to L. 274, which could support the role of soil N content in accumulating SOC.

Response: Thanks very much to the suggestions of the reviewers, we have revised the sentence.

Line 288 could be attributed to alluvial deposit…

Response: Response: Thanks very much to the suggestions of the reviewers, we have revised the sentence in line 300-303.

Line 296 soil inorganic C pool

Response: Thanks very much to the suggestions of the reviewers, we have revised the sentence.

Line 297-298 This concluded sentence is a bit contradictory with the previous sentence.

Response:The sentence structure description is incorrect. We will modify the high altitude area to the low altitude area.

Line 305 on the one hand

Response: Thanks very much to the suggestions of the reviewers, we have revised the sentence.

Line 311-313 Please modify this sentence more clearly

Response: Thanks very much to the suggestions of the reviewers, we have revised the sentence as "Our research results show that in the dry, high and cold plateau area of the Qinghai-Tibet Plateau, organic carbon increases linearly with the increase of altitude, while silicon carbide first increases and then decreases with the increase of altitude. "

---

## Author Comment (AC2)

**Reviewer #2:**

Thank you very much for the reviewers' suggestions. We have made the following modifications to the questions raised by the reviewers:

This manuscript investigates the altitudinal patterns of soil organic carbon (SOC) and inorganic carbon (SIC) in an arid alpine grassland ecosystem in the northern Qinghai-Tibetan Plateau. The study addresses a relevant and underexplored topic and contributes to our understanding of carbon pool dynamics in fragile, water- and temperature-constrained ecosystems. The finding that SOC increases with altitude while SIC shows a non-linear pattern is interesting and provides valuable contrast to previous studies in humid alpine zones. However, several issues related to English expression, logical structure, data analysis (particularly SEM), and the discussion of SIC need to be addressed before publication.

Specific comments:

1.The manuscript requires moderate editing for grammar and clarity. Some sentences are overly long or awkward (e.g., lines 233–235). Consider professional language polishing.

Response:According to the reviewers' comments, we made modifications to the long sentences in the article. For example, "In contrast to the findings from prior studies with alpine meadow or moist grasslands (Chen et al., 2017; Chen et al., 2022), our results show the predominance of soil inorganic C in the dry alpine rangeland." was changed to "Contrary to the previous research results that the soil carbon pool of alpine meadows or moist grasslands was mainly composed of organic carbon (Chen et al., 2017; Chen et al., 2022). Our research results show that inorganic carbon in the soil of arid and alpine plateaus dominates." in line 241-245.

2.The current version does not adequately explain the broader research background or the regional climate-vegetation features of the study area. Suggest briefly stating why arid alpine grasslands are important.

Response: Thanks very much to the suggestions of the reviewers, we have added the sentence about the importance of alpine grassland in arid areas in line 68-71. "Alpine

meadows in arid areas are also an important part of the alpine meadow ecosystem on the Qinghai-Tibet Plateau and are the foundation of agriculture and animal husbandry in the northern part of the Qinghai-Tibet Plateau (Zhang et al., 2021). ”

3.The description of climatic and vegetation features of the study area is insufficient in the introduction. Add 1–2 sentences summarizing key traits of arid alpine grasslands.

Response: Thanks very much to the suggestions of the reviewers, we have added the sentence about climatic and vegetation features in line 77-80. “ The alpine grassland in the northern part of the Qinghai-Tibet Plateau is located in the arid climate area, with a relatively low vegetation coverage. The surface is severely eroded by wind and water, and the community type is mainly desert grassland”.

4.Hypotheses lack specificity – In the introduction, the hypothesis should be expanded to state expected relationships between environmental, vegetation, and soil variables and SOC/SIC.

Response: Based on the reviewers' comments and the introduction in the preface, we will modify the hypothesis to (1) With the increase of altitude, the decrease of temperature and the increase of vegetation, the content of soil organic carbon increases; (2) With the increase of altitude, the drought limitation is alleviated and the inorganic carbon content in the soil decreases; (3) With the increase of altitude, the proportion of organic carbon in the soil carbon pool increases, while the proportion of inorganic carbon decreases.

5.line 94-95, 91°18'E, 36°N - 37°49'N, Punctuation error.

Response: Thanks very much to the suggestions of the reviewers, we have revised the sentence.

6.Line 105 – Figure S1 is cited but not provided.

Response: Change "Fig. S1" to "Fig. 1".

7.The description of the initial SEM model is missing. Please add information on model structure, variable selection rationale, and key fit indices (e.g., $\chi^2$, RMSEA).

Response: In this paper, the F-value test used in the SEM model calculation is used,

and the significance and fitting degree of the model are represented by Fisher's, $p$, AIC and BIC.

8.Line 259 – It is stated that MAT and SWC significantly increased SOCD and SICD, but no mechanistic interpretation is provided for SICD.

Response: Since this part of the content only explains the changes and influencing factors of SOCD, SICD is not mentioned. Therefore, we deleted "SICD".

9.Discussion on SIC / SIC:

The discussion of SIC is relatively fragmented. Recommend reorganizing it into two parts: (a) Abiotic controls (e.g., parent material, soil water, slope effects); (b) Potential biotic influences (e.g., root respiration, vegetation cover).

Response: According to the reviewers' comments, we analyzed the influencing factors of SICD changes at different altitude gradients into abiotic factors and biological factors. In line 298-314, we analyzed the influences of abiotic factors such as precipitation, topography, and soil parent material. In line 314-319, we analyzed the influence of biological factors such as plant communities. Finally, the main influencing factors and causes of this area were comprehensively analyzed through the SEM model analysis.

10.Line 303 – "SEM-composite" is unclear. Possibly rephrase as "composite SEM model" or "final SEM structure."

Response: Thanks very much to the suggestions of the reviewers, we changed "SEM-composite" to "composite SEM".

---

## Author Response (AR2)

**Editor comment:**

1. In your first reply it was sometimes unclear whether a given reviewer comment would be incorporated into the manuscript. Please prepare (a) a clean, fully revised manuscript, (b) a tracked-changes or annotated version, and (c) a point-by-point response letter that quotes each reviewer remark followed by your action or rebuttal and a page/line reference.

Response: After carefully revising the comments, we uploaded them to revised manuscript, annotated version and response letter.

2.Specifically, I would like to emphasize the need to elaborate the information about the SEM analysis. The current description is too brief to judge or reproduce the analysis. In the Methods, specify the SEM framework and estimator, the sample size and observational unit, the observed variables in each latent/composite block, and the full model-fit statistics together with the a-priori model diagram. In addition, outline the criteria for path pruning, document assumption checks, and provide the R code or model syntax plus the covariance/correlation matrix in the Supplement so that readers can replicate the analysis.

Response: According to the editor's comments, we have refined the information about SEM in the material method. We described the construction basis and screening criteria of the SEM model, and provided the original model, R code and correlation matrix in the supplement.

3.As noted previously, please replace altitude with elevation throughout the text, figures, and tables as well as in the title. As explained previously, 'elevation' is standard in terrestrial ecology and 'altitude' has a slightly different meaning.

Response: Thanks very much to the suggestions, we modified 'altitude' to 'elevation' in article.

**Reviewer #1:**

Thank you very much for the reviewers' suggestions. We have made the following modifications to the questions raised by the reviewers:

The manuscript by Liu et al. investigated the pattern of changes in soil organic carbon and soil inorganic carbon along the 3000-4000 m elevation range in the northern Qinghai-Tibetan Plateau region. In this study, SOCD increased with the elevation of elevation and was attributed to the changes of dominated vegetation cover and the decreasing MAT. SICD displayed a nonlinear change and was influenced by soil C/N ratio and soil water content. The topic of the study is very interesting and they found the divergent responses of soil C component to elevation. However, there still some issues needed to be addressed before its publication. First, it is important for the authors to present scientific questions more clearly based on summarizing previous research, and further improve the logical and English expression of the paper. Second, the figure and table notes in the result section need to be modified according to the format and need to be distinguished from the font in the manuscript. Third, in the discussion section, there is still a problem of logic ambiguity in the change of soil inorganic carbon density. Please re-organize the structure of this section, so as to make the logic of the discussion section more reasonable and clearly. Further, there are many long and complex sentences in the article, please modify wherever needed. I also have some specific comments listed below.

Response: Based on the reviewers' comments, we further clarified the assumptions, corrected the chart errors, and reorganized the analysis of SIC. The specific modification contents can be found in the main text of the article and specific comments.

Specific comments:

Line 15 Please describe the experimental design in detail

Response: In the abstract, we have added the description of the experimental design. "To address this uncertainty, we uniformly selected seven points on the vertical gradient at an elevation of 3,000-4,000 meters in the northern part of the Qinghai-Tibet Plateau to analyze the changing trends of organic carbon (SOC) and inorganic carbon (SIC) in the surface soil."

Line 27 Please add the "ratio" after C/N

Response: Thanks very much to the suggestions of the reviewers, we have revised the

sentence.

Line 32 Don't use abbreviations when writing keywords and try to spell the words in full.

Response: Thanks very much to the suggestions of the reviewers, we have revised the keywords.

Line 37 The meaning of the sentence is not clear, please revise it.

Response: We modify this sentence to "However, despite extensive studies on soil carbon pools in the past, there is still great uncertainty regarding the response of soil C pools to global climate change'.

Line 44: Please revised "whereas" to "Whereas"

Response: Thanks very much to the suggestions of the reviewers, we have revised the sentence.

Line 52 the driven factors?

Response: Based on the reviewers' opinions, we added specific influencing factors such as climate and vegetation. For specific modifications, please refer to line 52-54.

Line 53-55 This sentence is long, so please split into two short sentences.

Response: We modify this sentence to "The Qinghai-Tibet Plateau rises sharply in the elevation gradient, and the ecosystem types have the characteristics of spatial differentiation. Among different ecological types, there are significant differences in climate, vegetation and soil characteristics".

Line 58-63 Again, the description of research progress is also too long. It will be better to raise the knowledge gap in a separate sentence.

Response: According to the reviewers' comments, we modify this sentence to " However, most studies in this region have mainly focused on the vegetation or soil change patterns of permafrost and meadow ecosystems (Wang et al., 2023; Chen et al., 2017; Cai et al., 2025. The variation patterns of vegetation and soil carbon pools in alpine meadow ecosystems in arid areas have been largely ignored".

Line 70 delete most

Response: Thanks very much to the suggestions of the reviewers, we have revised the sentence.

Line 71 in the soil C storage

Response: Thanks very much to the suggestions of the reviewers, we have revised the sentence.

Line 72 Previous studies may have neglected the role of SIC in soil carbon inventory in this region.

Response: Thanks very much to the suggestions of the reviewers, we added this sentence.

Line 73-75 Please delete this repetitive sentence.

Response: Thanks very much to the suggestions of the reviewers, we have revised the sentence.

Line 75-78 These statement about the elevation changes in vegetation and soil nutrient availability should have appeared earlier in this paragraph.

Response: We placed the description of the changes in vegetation and nutrients on the elevation gradient in paragraphs line 58-62.

Line 79 whether or to what extent the altitudinal changes in micro-environments and vegetation would affect SOC and SIC remains unknown.

Response: Thanks very much to the suggestions of the reviewers, we have revised the sentence.

Line 87-90 The two hypotheses are similar: the relative importance of SOC to SIC decreases with elevation vs. SIC would become more profound at higher elevations.

Response: According to the reviewers' comments, we modified the hypothesis as "(1) With the increase of elevation, the decrease of temperature and the increase of vegetation, the content of soil organic carbon increases; (2) With the increase of elevation, the drought limitation is alleviated and the inorganic carbon content in the soil decreases; (3) With the increase of elevation, the proportion of organic carbon in the soil carbon pool increases, while the proportion of inorganic carbon decreases."

Line 115 The description of the methods that the experiment was conducted should be further addressed.

Response: Thanks very much to the suggestions of the reviewers, we have revised the methods in line 118-121.

Line 126 The soil was collected from 0-30 cm, which was different from 0-20 cm in other literatures. Please explain in the materials and methods and explain why this soil layer is used.

Response: This is mainly because in this study area, there are more vegetation such as dwarf shrubs, and the root systems are mostly distributed in the range of 0-30cm. Therefore, we chose a soil layer of 0-30cm.

Line 136 The measuring method of SIC is unclear. Is it a calculated value by subtracting SOC from TC or a measured value?

Response: The content of SIC was directly determined by neutralization titration.

Fig. 2b There were two regression lines, but only one R2 and P values was reported?

Line 198 In Table 1 and Figs 3, 4, the listed variables mismatched, please revise them in the result.

Response: We have corrected the errors in the chart. For the specific correction content, please refer to the chart in the main text.

Line 235: showed

Response: Thanks very much to the suggestions of the reviewers, we have revised the sentence.

Line 236-238: Further, soil organic C pool linearly increased, while soil inorganic C pool appears to be nonlinear along the altitudinal gradient.

Response: Thanks very much to the suggestions of the reviewers, we have added the sentence in line 246-247.

Line 246 Unexpectedly, this study found the negative correlation between soil organic C pool and aboveground biomass density. This could be explained by…

Response: Thanks very much to the suggestions of the reviewers, we have added the sentence in line 255-257.

Line 258 showed

Response: Thanks very much to the suggestions of the reviewers, we have revised the sentence.

Line 263, 264 delete "effect"

Response: Thanks very much to the suggestions of the reviewers, we have revised the

sentence.

Line 267-271 Please make a summarized conclusion about how vegetation abundance and climate together affect SOC here.

Response: According to the research content, a summary of the impact on SOC has been added at the end of this paragraph. "Therefore, with the increase of the elevation gradient and the decrease of temperature, the vegetation type gradually changes from desert to grassland, the soil carbon input increases, the mineralization rate decreases, and the SOC content increases."

Line 278 Under this condition, plants may grow faster…

Response: Thanks very much to the suggestions of the reviewers, we have added the sentence in line 291.

Line 280-282 Please move this main result to L. 274, which could support the role of soil N content in accumulating SOC.

Response: Thanks very much to the suggestions of the reviewers, we have revised the sentence.

Line 288 could be attributed to alluvial deposit…

Response: Response: Thanks very much to the suggestions of the reviewers, we have revised the sentence in line 300-303.

Line 296 soil inorganic C pool

Response: Thanks very much to the suggestions of the reviewers, we have revised the sentence.

Line 297-298 This concluded sentence is a bit contradictory with the previous sentence.

Response:The sentence structure description is incorrect. We will modify the high elevation area to the low elevation area.

Line 305 on the one hand

Response: Thanks very much to the suggestions of the reviewers, we have revised the sentence.

Line 311-313 Please modify this sentence more clearly

Response: Thanks very much to the suggestions of the reviewers, we have revised the

sentence as "Our research results show that in the dry, high and cold plateau area of the Qinghai-Tibet Plateau, organic carbon increases linearly with the increase of elevation, while silicon carbide first increases and then decreases with the increase of elevation. "

**Reviewer #2:**

Thank you very much for the reviewers' suggestions. We have made the following modifications to the questions raised by the reviewers:

This manuscript investigates the altitudinal patterns of soil organic carbon (SOC) and inorganic carbon (SIC) in an arid alpine grassland ecosystem in the northern Qinghai-Tibetan Plateau. The study addresses a relevant and underexplored topic and contributes to our understanding of carbon pool dynamics in fragile, water- and temperature-constrained ecosystems. The finding that SOC increases with elevation while SIC shows a non-linear pattern is interesting and provides valuable contrast to previous studies in humid alpine zones. However, several issues related to English expression, logical structure, data analysis (particularly SEM), and the discussion of SIC need to be addressed before publication.

Specific comments:

1.The manuscript requires moderate editing for grammar and clarity. Some sentences are overly long or awkward (e.g., lines 233–235). Consider professional language polishing.

Response:According to the reviewers' comments, we made modifications to the long sentences in the article. For example, "In contrast to the findings from prior studies with alpine meadow or moist grasslands (Chen et al., 2017; Chen et al., 2022), our results show the predominance of soil inorganic C in the dry alpine rangeland." was changed to "Contrary to the previous research results that the soil carbon pool of alpine meadows or moist grasslands was mainly composed of organic carbon (Chen et al., 2017; Chen et al., 2022). Our research results show that inorganic carbon in the soil of arid and alpine plateaus dominates." in line 241-245.

2.The current version does not adequately explain the broader research background or the regional climate-vegetation features of the study area. Suggest briefly stating why arid alpine grasslands are important.

Response: Thanks very much to the suggestions of the reviewers, we have added the sentence about the importance of alpine grassland in arid areas in line 68-71. "Alpine meadows in arid areas are also an important part of the alpine meadow ecosystem on the Qinghai-Tibet Plateau and are the foundation of agriculture and animal husbandry in the northern part of the Qinghai-Tibet Plateau (Zhang et al., 2021). "

3.The description of climatic and vegetation features of the study area is insufficient in the introduction. Add 1–2 sentences summarizing key traits of arid alpine grasslands.

Response: Thanks very much to the suggestions of the reviewers, we have added the sentence about climatic and vegetation features in line 77-80. " The alpine grassland in the northern part of the Qinghai-Tibet Plateau is located in the arid climate area, with a relatively low vegetation coverage. The surface is severely eroded by wind and water, and the community type is mainly desert grassland".

4.Hypotheses lack specificity – In the introduction, the hypothesis should be expanded to state expected relationships between environmental, vegetation, and soil variables and SOC/SIC.

Response: Based on the reviewers' comments and the introduction in the preface, we will modify the hypothesis to (1) With the increase of elevation, the decrease of temperature and the increase of vegetation, the content of soil organic carbon increases; (2) With the increase of elevation, the drought limitation is alleviated and the inorganic carbon content in the soil decreases; (3) With the increase of elevation, the proportion of organic carbon in the soil carbon pool increases, while the proportion of inorganic carbon decreases.

5.line 94-95, 91°18'E, 36°N - 37°49'N, Punctuation error.

Response: Thanks very much to the suggestions of the reviewers, we have revised the sentence.

6.Line 105 – Figure S1 is cited but not provided.

Response: Change "Fig. S1" to "Fig. 1".

7.The description of the initial SEM model is missing. Please add information on model structure, variable selection rationale, and key fit indices (e.g., $\chi^2$, RMSEA).

Response: In this paper, the F-value test used in the SEM model calculation is used, and the significance and fitting degree of the model are represented by Fisher's, $p$, AIC and BIC.

8.Line 259 – It is stated that MAT and SWC significantly increased SOCD and SICD, but no mechanistic interpretation is provided for SICD.

Response: Since this part of the content only explains the changes and influencing factors of SOCD, SICD is not mentioned. Therefore, we deleted "SICD".

9.Discussion on SIC / SIC:

The discussion of SIC is relatively fragmented. Recommend reorganizing it into two parts: (a) Abiotic controls (e.g., parent material, soil water, slope effects); (b) Potential biotic influences (e.g., root respiration, vegetation cover).

Response: According to the reviewers' comments, we analyzed the influencing factors of SICD changes at different elevation gradients into abiotic factors and biological factors. In line 298-314, we analyzed the influences of abiotic factors such as precipitation, topography, and soil parent material. In line 314-319, we analyzed the influence of biological factors such as plant communities. Finally, the main influencing factors and causes of this area were comprehensively analyzed through the SEM model analysis.

10.Line 303 – "SEM-composite" is unclear. Possibly rephrase as "composite SEM model" or "final SEM structure."

Response: Thanks very much to the suggestions of the reviewers, we changed "SEM-composite" to "composite SEM".

**References**

Cai, W. X., Xu, L., Wen, D., Zhou, Z. Y., Li, M. X., Wang, T., & He, N. P. The carbon sequestration potential of vegetation over the Tibetan Plateau. Renewable & Sustainable Energy Reviews, 207, 114937. https://doi.org/ARTN

11493710.1016/j.rser.2024.114937, 2025.

Chen, L. T., Jing, X., Flynn, D. F. B., Shi, Y., Kühn, P., Scholten, T., & He, J. S. Changes of carbon stocks in alpine grassland soils from 2002 to 2011 on the Tibetan Plateau and their climatic causes. Geoderma, 288, 166-174. https://doi.org/10.1016/j.geoderma.2016.11.016, 2017.

Chen, Y., Han, M. G., Yuan, X., Hou, Y. H., Qin, W. K., Zhou, H. K., Zhao, X. Q., Klein, J. A., & Zhu, B. A. Warming has a minor effect on surface soil organic carbon in alpine meadow ecosystems on the Qinghai-Tibetan Plateau. Global Change Biology, 28, 1618-1629. https://doi.org/10.1111/gcb.15984, 2022.

Li, J., Pei, J., Fang, C., Li, B., & Nie, M. Drought may exacerbate dryland soil inorganic carbon loss under warming climate conditions. Nature Communications, 15, 617. https://doi.org/10.1038/s41467-024-44895-y, 2024.

Mi, N. A., Wang, S., Liu, J., Yu, G., Zhang, W., & Jobbagy, E. Soil inorganic carbon storage pattern in China. Global change biology, 14, 2380-2387.. https://doi.org/10.1111/j.1365-2486.2008.01642.x, 2008.

Wang, Z. W., Huang, L. M., & Shao, M. A. Spatial variations and influencing factors of soil organic carbon under different land use types in the alpine region of Qinghai-Tibet Plateau. Catena, 220, 106706. https://doi.org/ARTN 10670610.1016/j.catena.2022.106706, 2023.

Zhang, A., Li, X., Wu, S., Li, L., Jiang, Y., Wang, R., ... & Li, L. Spatial pattern of C: N: P stoichiometry characteristics of alpine grassland in the Altunshan Nature Reserve at North Qinghai-Tibet Plateau. Catena, 207, 105691. https://doi.org/10.1016/j.catena.2021.105691, 2021.

---

## Author Response (AR3)

Thank you for your revised manuscript, which has been re-evaluated by the original reviewers. Both reviewers were generally satisfied with the changes that were made. One reviewer provided specific editing suggestions to further improve the text.

I also strongly recommend to carefully check the text for language improvement. In several places, the phrasing is awkward.

For example, l. 259 ff: "Contrary to the previous research results that the soil carbon pool of alpine meadows or moist grasslands was mainly composed of organic carbon (Chen et al., 2017; Chen et al., 262 2022). Our research results show that inorganic carbon in the soil of arid and alpine263 plateaus dominates." This is linguistically incorrect and should be rephrased. For example like this: Contrary to previous studies reporting that the soil carbon pool of alpine meadows and moist grasslands is predominantly organic (Chen et al., 2017; Chen et al., 2022), our findings indicate that inorganic carbon dominates in the soils of arid, alpine plateaus.

In addition, please replace "altitude" and "altitudinal" throughout the manuscript. This is the 3rd time I am asking this. The current manuscript has a mixture of terminology, as altitude/altitudinal was replaced in some places but not in all. This is confusing.

Response: We extend our sincere gratitude to the editors and reviewers for their valuable comments. In response to the concerns regarding inaccurate word usage, we have undertaken a thorough professional polishing and meticulous revision of the wording and sentences. During this process, we also addressed and corrected any incorrect or inaccurate descriptions within the manuscript. For instance, to better align with the research on the carbon pools of SIC and SOC, we modified "(1) With the increase of elevation, the decrease of temperature and the increase of vegetation, the content of soil organic carbon increases; (2) With the increase of elevation, the drought limitation is alleviated and the inorganic carbon content in the soil decreases; (3) With the increase of elevation, the proportion of organic carbon in the soil carbon pool increases, while the proportion of inorganic carbon decreases" to "(1) with

decreases in temperature along the elevational gradient, slow-turnover woody plants give way to fast-turnover herbaceous plants due to energy constraints, leading to greater SOC preservation at the higher elevational sites as a result of dual-mechanism of greater plant-derived C inputs and lower rate of decomposition; and (2) SIC dominates the soil C pool and would not display an apparent trend of variations with elevation as it is predominantly determined by soil parent materials and influenced by abiotic factors" for greater precision. Additionally, we carefully reviewed the entire content and replaced or modified all instances of "altitude" and "altitudinal". We apologize for the previous errors. For detailed modifications, please refer to the revised manuscript.